# Structural dynamics in proteins induced by and probed with X-ray free-electron laser pulses

Karol Nass [1,9], Alexander Gorel [1,9], Malik M. Abdullah [2,3], Andrew V. Martin[4], Marco Kloos[1], Agostino Marinelli [5], Andrew Aquila[5], Thomas R. M. Barends [1], Franz-Josef Decker[5], R. Bruce Doak[1], Lutz Foucar[1], Elisabeth Hartmann[1], Mario Hilpert [1], Mark S. Hunter[5], Zoltan Jurek[2,3], Jason E. Koglin[5], Alexander Kozlov[6], Alberto A. Lutman [5], Gabriela Nass Kovacs[1], Christopher M. Roome[1], Robert L. Shoeman [1], Robin Santra [2,3,7], Harry M. Quiney[6✉], Beata Ziaja[2,3,8✉], Sébastien Boutet [5] & Ilme Schlichting [1✉]

X-ray free-electron lasers (XFELs) enable crystallographic structure determination beyond the limitations imposed upon synchrotron measurements by radiation damage. The need for very short XFEL pulses is relieved through gating of Bragg diffraction by loss of crystalline order as damage progresses, but not if ionization events are spatially non-uniform due to underlying elemental distributions, as in biological samples. Indeed, correlated movements of iron and sulfur ions were observed in XFEL-irradiated ferredoxin microcrystals using unusually long pulses of 80 fs. Here, we report a femtosecond time-resolved X-ray pump/X-ray probe experiment on protein nanocrystals. We observe changes in the protein backbone and aromatic residues as well as disulfide bridges. Simulations show that the latter's correlated structural dynamics are much slower than expected for the predicted high atomic charge states due to significant impact of ion caging and plasma electron screening. This indicates that dense-environment effects can strongly affect local radiation damage-induced structural dynamics.

[1] Max-Planck-Institut für Medizinische Forschung, Jahnstraße 29, 69120 Heidelberg, Germany. [2] Center for Free-Electron Laser Science, Deutsches Elektronen-Synchrotron DESY, Notkestrasse 85, 22607 Hamburg, Germany. [3] The Hamburg Centre for Ultrafast Imaging, Luruper Chaussee 149, 22761 Hamburg, Germany. [4] School of Science, RMIT University, 124 La Trobe Street, Melbourne, VIC 3000, Australia. [5] SLAC National Accelerator Laboratory, Menlo Park, CA 94025, USA. [6] ARC Centre of Excellence for Advanced Molecular Imaging, School of Physics, The University of Melbourne, Melbourne, VIC 3010, Australia. [7] Department of Physics, Universität Hamburg, Jungiusstrasse 9, 20355 Hamburg, Germany. [8] Institute of Nuclear Physics, Polish Academy of Sciences, Radzikowskiego 152, 31-342 Kraków, Poland. [9]These authors contributed equally: Karol Nass, Alexander Gorel. ✉email: beata.ziaja-motyka@cfel.de; quiney@unimelb.edu.au; ilme.schlichting@mpimf-heidelberg.mpg.de

The intense femtosecond X-ray pulses of XFELs enable breakthrough science, spanning a wide range of scientific fields and approaches. These include femtochemistry, probing the structural dynamics upon photoexcitation at high spatial and temporal resolution[1,2] and allowing the capture of molecular movies of chemical reactions[3], as well as the exploration of extreme physical regimes of X-ray-matter interaction[4]. Interestingly, these two themes are entwined: interaction with a focused XFEL beam inevitably creates an extremely non-equilibrium state of matter before the pulse terminates. Pulse length thus not only matters but, according to predictions[5], must be as short as 10 fs to allow protein structure determination[6]. Based on recent successes[7–9], however, confidence has been building that atomic structure and dynamics can still be probed in an essentially 'damage-free' manner via serial femtosecond crystallography (SFX)[10] even when using much longer XFEL pulses. This is attributed to a 'self-gating' of Bragg diffraction by loss of crystalline order upon ionization, such that 'effective' pulse lengths that are significantly shorter than the actual pulse duration[2]. However, self-gating reflects an average damage over all atoms in the crystal ('global damage') and ignores distinct specific elemental distributions or the effects of the molecular environment on the local motion of ions, both of which can yield 'local damage' even before overall crystalline order is lost[11,12].

Indeed, there is increasing evidence that local X-ray-induced dynamics prevail in femtosecond XFEL experiments: Gas-phase molecule studies involving heavy elements have shown molecular ionization effects and charge rearrangement[13–15]. Similar outcomes are expected in dense systems. In inorganic crystals, coherent electronic rearrangement in $C_{60}$ has been observed[16] and likewise a transient lattice contraction in the solid-to-plasma transition of xenon clusters[17]. In protein crystals, the existence of local damage was inferred from data statistics early on[11], before distinct correlated dynamics in iron-sulfur clusters in ferredoxin microcrystals were observed experimentally[18] and reproduced by simulations[19]. The observation of correlated motions of atoms, in particular of those close to heavy atoms[18–20], is of great concern in structural biology as it is exactly the knowledge of their precise coordination that is needed to deduce the catalytic mechanism of e.g., metalloenzymes. Indeed, recent simulations suggest an elongation of the μ-oxo bridge (O5) in the photosystem II oxygen-evolving-complex due to radiation damage inflicted by the XFEL pulse[21]; other simulations have demonstrated enhanced ionization of light atoms bound to heavy ones in molecules[14]. It is therefore important to understand how the local dynamics in soft condensed matter, including biological molecules, are dominated by local molecular or atomic configuration and especially by the plasma-like state created by the XFEL pulse[19]. To this end, there is a need for experimental observations of local atomic dynamics in molecules combined with consistent theoretical models to reveal and explain the details of molecular ionization in XFEL experiments. Such an understanding is an essential foundation for studies using XFEL sources for the determination of chemically meaningful structures.

Here, we describe a time-resolved X-ray pump, X-ray probe SFX experiment that we performed at the Linac Coherent Light Source (LCLS) to monitor structural changes induced in protein nanocrystals by an XFEL pulse. We studied thaumatin and a gadolinium complex of lysozyme[22]. The latter allows the study of the evolution of electronic damage in heavy atoms, which is of practical interest for radiation damage-induced phasing schemes[23,24]. To separate the diffraction patterns of the pump and probe X-ray pulses (1 mJ pulse energy, split roughly equally between the two 15 fs long pulses, each of $3.5 \times 10^{12}$ photons μm$^{-2}$ fluence, corresponding to an average intensity of $2.7 \times 10^{19}$ W cm$^{-2}$ in the focus (nominally ~0.2 μm FWHM, see

Supplementary Discussion), their photon energies were chosen to lie above and below the iron K-absorption edge (~7.11 keV), respectively. A thin iron foil in front of the detector absorbed the pump but not the probe pulse[17] (Supplementary Fig. 1). In addition to collecting X-ray pump X-ray probe SFX data, we also collected diffraction data using a single pulse. The experimental measurements were complemented with two separate theoretical approaches to follow X-ray induced ionization and local structural dynamics in detail in the protein samples.

## Results and discussions

**Experimental results.** The temporal evolution of radiation damage induced by XFEL pulses in lysozyme crystals has been studied previously at the LCLS using a split and delay setup to vary the time delay between pump and probe pulse between 19 and 213 fs[25]. The macroscopic crystals were mounted on chips, and diffraction from pump and probe pulse, respectively, was spatially separated on the detector using gratings. The geometry of the setup limited the resolution of the diffraction data to 8 Å, preventing analysis of local damage and complicated the scaling of the intensities of the few Bragg peaks captured on the detector. Nevertheless, the authors concluded that there are no indications for radiation damage which they attributed to the low flux and thus dose[25]. In contrast, in our experimental design the accelerator was used to generate two pulses with different photon energies, with the accelerator controlling the delay between them. The photon energy difference allows an absorbing foil to remove the pump pulse before reaching the detector. This enabled acquisition of high-resolution SFX data permitting the analysis of local and global damage.

During the 15 fs pump X-ray pulse, the thaumatin and lysozyme.Gd nanocrystals absorbed a dose of 7 GGy[26] and 33 GGy[26] (omitting photoelectron escape effects), respectively. Despite this, the nanocrystals diffracted to better than 2.0 Å resolution. Due to shadowing of the X-ray detector by the injector shroud, we truncated the data at 2.32 Å resolution (Supplementary Fig. 2). We aimed for similar pulse energies for the X-ray pump and probe pulses and equally spaced time delays, increasing from 20 fs to 100 fs in 15–20 fs steps. However, due to very limited time for tuning of the accelerator neither goal was consistently met. Moreover, machine performance likely differed slightly during the shifts of the experiment (see Supplementary Table 1 for the experimentally derived values). This may explain inconsistencies in magnitude of the displacements observed in lysozyme.Gd for the nominally 20 fs and 40 fs time delays (determined to be 35 and 37 fs, respectively, see Supplementary Note 2).

The nanocrystals continued to diffract to high resolution even at the longest time delay of nominally 100 fs. However, the data quality decreases with longer pump probe time delays as indicated by worsening of the data quality indicators (R$_{split}$, CC, I/σ(I)), as well as by the increase of the Wilson B-factor and decrease of I/σ(I) (Supplementary Figs. 2 and 3), indicating global radiation damage (Supplementary Tables 1 and 2). The statistics (such as R$_{split}$, CC and I/σ(I)) of the single-pulse data are worse than that of the double-pulse data at short time delays, most likely due to the significantly lower pulse energy. In contrast to thaumatin, the cumulative intensity distributions of the lysozyme.Gd data differed from the expected values. For the single pulse and the nominal 20 fs time delay data, the distributions are normal, but for increasing time delays the fraction of weak reflections changed (Supplementary Fig. 4). This may be due to increased radiation damage in the presence of 100 mM Gd in the mother liquor, increasing the concentration of highly ionized atoms and thus dose.

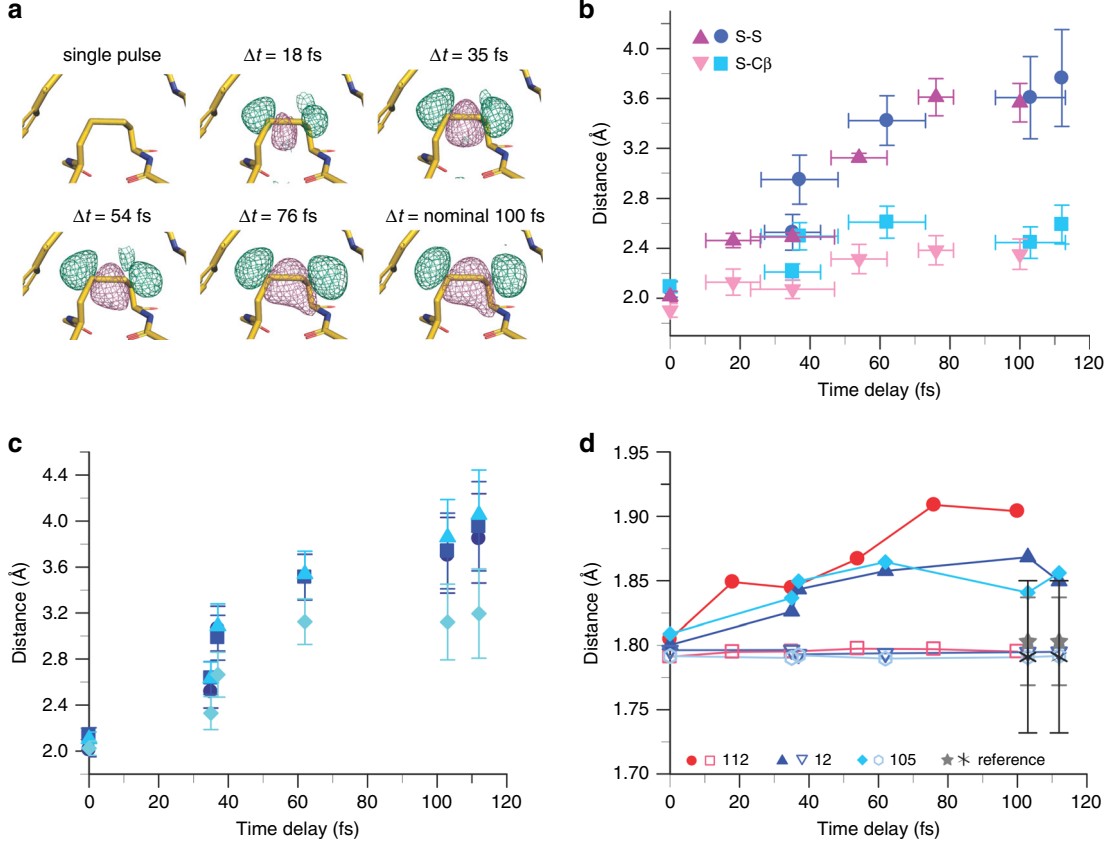

**Fig. 1 Disulfide and thioether bonds. a** Isomorphous difference density maps ($F_{obs(\Delta t)} - F_{obs(single\ pulse)}$) of thaumatin as a function of pump probe delay show negative (pink) peaks between the exemplarily chosen Cys56 and Cy66 bond and positive (green) peaks on the outside of the bond (contour level ± 3σ). This is consistent with an elongation of the S–S bonds. The other disulfide bonds in thaumatin and lysozyme.Gd show similar effects (Supplementary Figs. 7 and 8). **b** The average distance of the two sulfurs and the cysteine Cβ and sulfur Sγ in the disulfide bonds in thaumatin (8 S–S bonds, red symbols) and lysozyme.Gd (4 S–S bonds, blue symbols) increases with the time delay between pump and probe pulse. The error bars (y-axis) give the standard deviation of the distribution; they do not represent the accuracy of the distance determination. The errors in pump probe delays (x-axis) are the standard deviations of the XTCAV-derived time-delay values. **c** The extent of bond elongation and the direction of the movement of the sulfur atoms depends on their local environment (Supplementary Note 2, Supplementary Fig. 10–12). In lysozyme.Gd the elongation of the Cys76-Cys94 bond (cyan diamonds) differs from that of the other three S–S bonds (Cys6-Cys127 (blue circles), Cys30-Cys115 (blue squares), Cys64-Cys80 (blue triangles)). The trajectories and local environment of the moving sulfur ions is shown in Supplementary Fig. 11. **d** Methionine residues. The Cγ-Sδ bond (filled symbols) in methionine residues lengthens significantly with pump probe delay in both thaumatin (red) and lysozyme.Gd (blue). The numbers correspond to the sequence number in the protein. The sulfur-Cε-methyl moiety seems to move as an entity given the apparent invariance of the Cε-S bond (open symbols). Reference bond lengths[60] (Cγ-Sδ (filled gray stars), Cε-Sδ (gray stars)) and their standard deviations are shown in gray.

In addition to an overall data quality decrease with delay time (Supplementary Tables 1 and 2, Supplementary Figs. 2–4), consistent with Bragg termination (see Supplementary Note 2, Supplementary Fig. 5) and a general deterioration of the electron density maps, we observed distinct structural changes with time. The integrity of disulfide (S–S) bonds (Supplementary Fig. 6) is often considered a marker for local radiation damage[20,27,28]. Lysozyme contains four S–S bonds, thaumatin eight. To visualize the evolution of structural changes with time delay we calculated isomorphous difference maps ($F_{obs(\Delta t)} - F_{obs(single\ pulse)}$[29]) for thaumatin and lysozyme.Gd. In both cases there is strong positive and negative electron density around the disulfide bridges indicative of a lengthening of the S–S distance with pump probe time delay (Fig. 1, Supplementary Figs. 7, 8, 10). The overall magnitude, speed (~1000 m s⁻¹) and kinetics of the average S–S bond elongation are very similar for both lysozyme. Gd and thaumatin despite the very different structures and dose (Fig. 1). Nonetheless, the individual displacements differ and depend on the local environment. As expected, the trajectories of the sulfur ions correlate with closeness to nearby residues;

strikingly, however, the trajectories of sulfurs pointing towards the solvent decelerate to a halt (Supplementary Figs. 10 and 11). A detailed description, analysis and discussion of the sulfur ion trajectories can be found in the Supplementary Notes 1, 2. In addition to bonds involving the relatively heavy sulfur atoms in cysteine and methionine side chains (Fig. 1), we also observe distinct structural changes in the peptide backbone (Fig. 2, Supplementary Fig. 13). Positive difference electron density is apparent in the isomorphous difference maps $F_{obs(\Delta t)} - F_{obs(single\ pulse)}$[29] close to the carbonyl oxygen atoms and away from the peptide bond, suggestive of a bond elongation (Fig. 2). Interestingly, while the carbonyl C–O and N–C$_{alpha}$ bond lengths increase significantly, this effect is barely noticeable for the N–C and C–C$_{alpha}$ bond (Fig. 2d, Supplementary Fig. 13a). The effect is strongest for the carbonyl C–O bond length for the 18 fs time delay in thaumatin (Fig. 2e). The overall deterioration in data quality at longer time delays does not allow distinguishing whether the decrease in bond length for longer pump probe delays is real (see Supplementary Note 2). Notably, the electron densities of aromatic side chains also change with pump probe

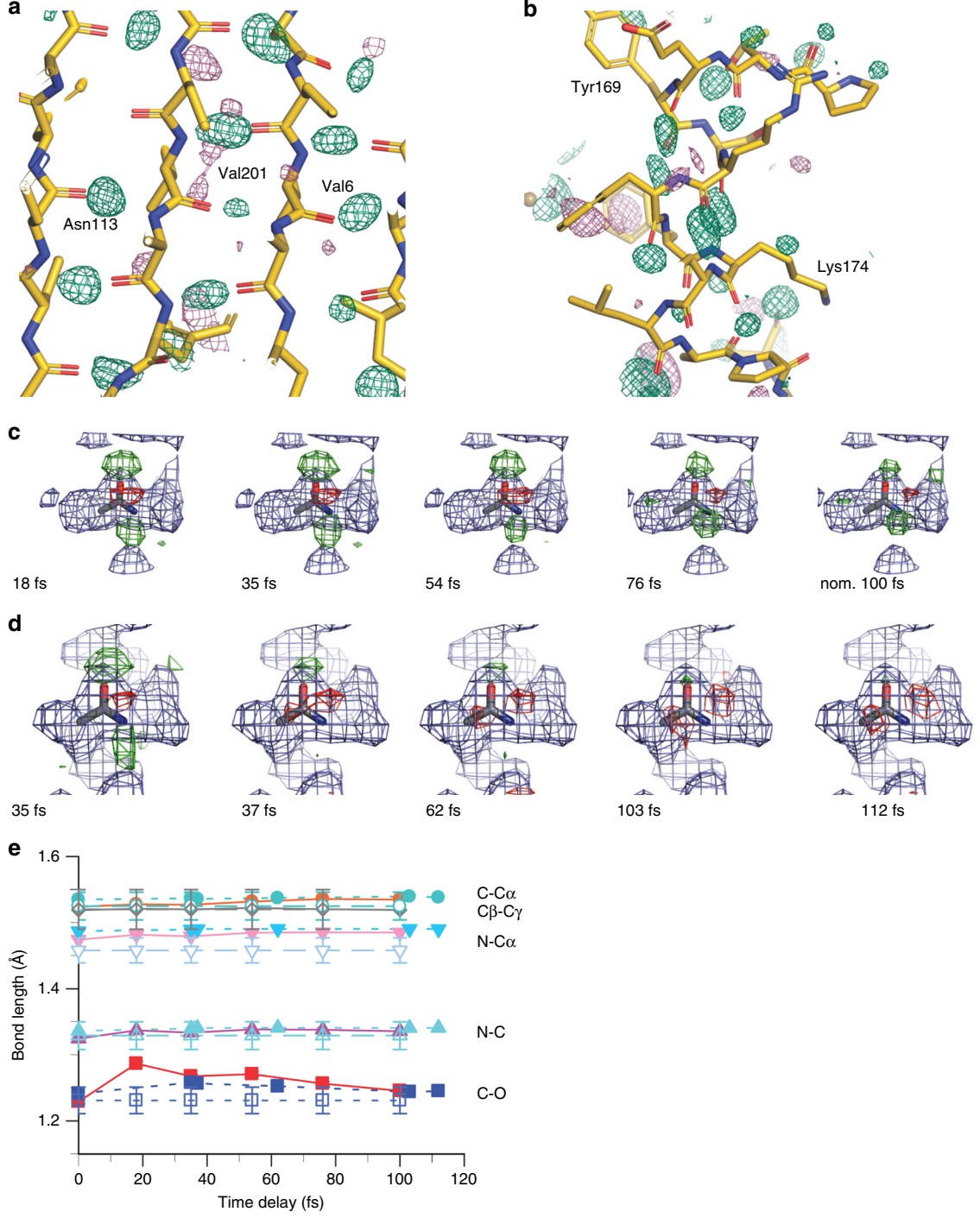

**Fig. 2 Changes in the protein backbone. a, b** Isomorphous difference density map ($F_{obs(18fs)} − F_{obs(single\ pulse)}$)[29] of thaumatin at a contour level of +3σ (green) shows peaks close to carbonyl oxygen atoms involved in hydrogen bonds in the ß-sheet region (**a**) and the α-helix region (**b**). There are fewer negative (−3σ pink) than positive (+3σ green) difference peaks. **c, d** Isomorphous difference density maps ($F_{obs(Δt)}−F_{obs(single\ pulse)}$) of thaumatin (**c**) and lysozyme.Gd (**d**) averaged over all peptide bonds shows also negative peaks (−3σ (red) and +3σ (green)). Both proteins show the effect, but it is less dependent on the delay time in case of lysozyme.Gd. This may be due to data quality; the lysozyme.Gd data deteriorate much faster than those of thaumatin (Supplementary Figs. 2–4, Supplementary Tables 1 and 2). **e** Refined bond lengths of the peptide bonds in lysozyme.Gd (blue filled symbols) and thaumatin (red filled symbols). The bond lengths are average values of 100 independently refined structures using a jackknife approach. The values of standard bond lengths[60] are displayed using open symbols.

delay, there are distinct negative peaks in the center of the rings in the isomorphous difference maps $F_{obs(Δt)} − F_{obs(single\ pulse)}$[29] (Fig. 3). In contrast, the ratio of the electron densities of the Gd ions and light atoms hardly changes with pulse delay (see also[24], Supplementary Fig. 14).

**Theoretical analysis.** To obtain insight into the X-ray pulse induced molecular processes, we simulated the dynamics of the ions in irradiated protein samples, focusing on the behavior of S–S bonds. To ensure that the conclusions are not biased by a particular model implementation, the sulfur ion dynamics were

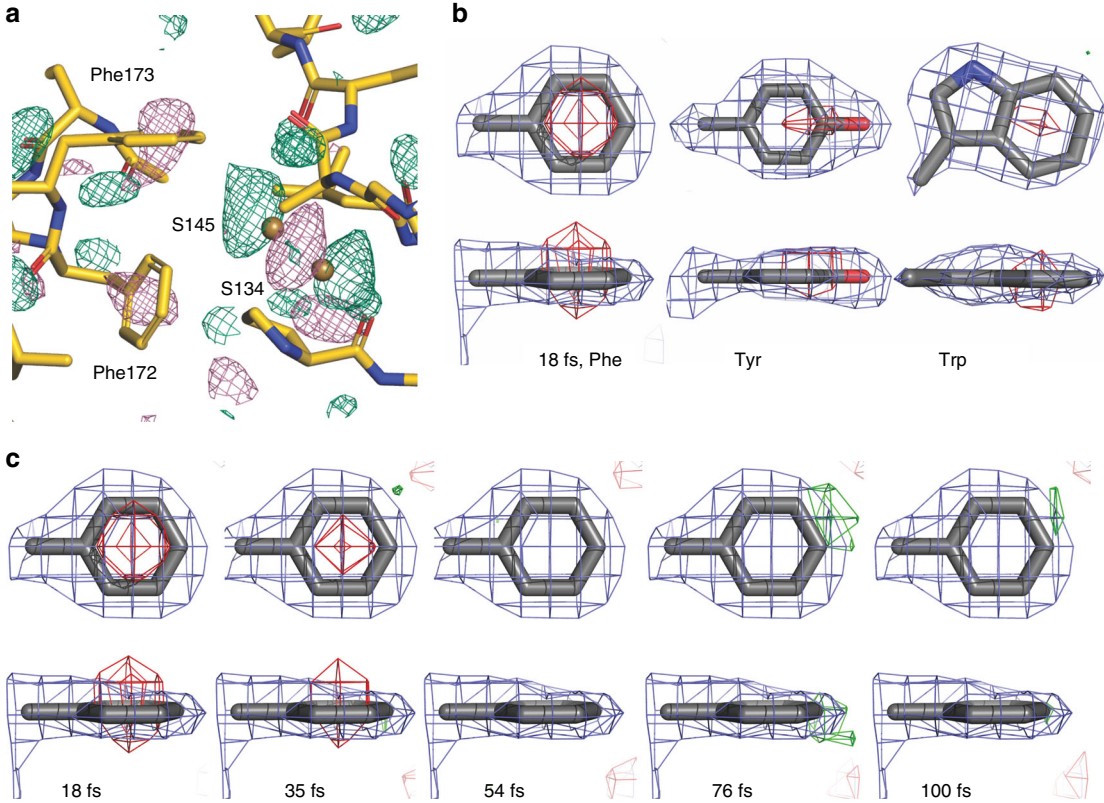

**Fig. 3 Changes in aromatic side chains. a** The isomorphous difference electron density map ($F_{obs(18fs)} - F_{obs(single pulse)}$)[29] of thaumatin shows peaks ($-3\sigma$ (pink) and $+3\sigma$ (green)) in the center of phenyl rings. In addition, there are changes around the adjacent elongated Cys145-Csy134 disulfide bridge and the backbone carbonyl oxygen atoms. **b** Averaged difference density (18 fs time delay point) of all phenylalanines (11), tyrosines (8) and tryptophan (3) side chain in thaumatin. **c** The difference electron density maps averaged over all phenylalanine residues shows that the negative difference is highest after 18 fs and no longer visible at 54 fs. It is unclear whether this latter observation is an effect of data quality.

treated by modeling the trapped electrons both as classical point particles and as a gaseous continuum. In the former case, a non-equilibrium molecular dynamics (MD) study of ion dynamics in thaumatin was performed with the XMDYN package[30–32] at three different X-ray fluences (see Methods section). The simulations show that due to the intense irradiation, the thaumatin sample becomes strongly ionized. Sulfur atoms, being the heaviest atomic constituent, reach the highest charge states of up to +5 (see Supplementary Fig. 15). For all cases of modeled values of fluence (F), the predicted S–S separation distances are comparable to the experimental values (Fig. 4a). In particular, for the calculations for the $F_{low}$ fluence case ($8.8 \times 10^{11}$ photons μm$^{-2}$) the S–S distance increases from 2.08 Å to ~3.3 Å, whereas for the $F_{med}$ and $F_{max}$ cases ($4.4 \times 10^{12}$ photons μm$^{-2}$ and $7.0 \times 10^{12}$ photons μm$^{-2}$, respectively) the mean displacements (as a function of time delay) do not exceed ~4.7 Å. For comparison, in vacuum, even at the lowest fluence considered, the S–S distance can reach up to 8 Å (see Supplementary Fig. 16). Increasing the fluence in vacuum by a factor of 10 enhances the relative S–S displacement 5–8 times. In contrast, within the dense protein environment the same change of fluence by a factor of 10 increases the S–S distance only by a factor of 1.4. This shows that the dissociation of the disulfide bridge in protein samples is strongly influenced by both the high charge of its atomic constituents and by the charged environment of the bridge consisting of free electrons and non-S ions (Supplementary Fig. 16). This environment significantly slows the S–S separation and induces a complex motion of the bridge involving both translational and vibrational modes (Supplementary Fig. 17), in marked contrast to the disulfide behavior in vacuum (Supplementary Fig. 16), where

the center-of-mass motion and the relative motion of the S ions are uncoupled. Further, for the $F_{med}$ and $F_{max}$ fluence cases, one can observe a plateau feature at time delays between 20 fs and 40 fs, arising through Coulomb repulsion of the separating S ions by the nearest neighboring non-S ions (Supplementary Fig. 16e, f and its discussion). This 'ion caging' effect transiently slows down the dissociation of the disulfide bridge.

We obtained a complementary insight into the physics of trapped electrons using a continuum electron plasma model. Within this hybrid approach, only the ion trajectories are explicitly calculated. The plasma screening of ion–ion Coulomb interactions by the unbound electrons is modeled with pairwise potentials[33] (see Methods). A simulation of ion dynamics in lysozyme with the continuum treatment (Fig. 4b) reveals that the S–S Coulomb interaction is almost entirely 'screened' by trapped electrons beyond a time delay of 60 fs when the S–S distance exceeds the transient Debye length. Approaching a 100 fs time delay, interactions with environmental ions ('ion caging') further limit the motion of the sulfur ions. In summary, the analysis of two different model systems using either the XMDYN molecular dynamics approach or the continuum model consistently shows that the experimentally observed S–S deceleration is due to ion caging and plasma electrons effects.

In conclusion, using a femtosecond time-resolved X-ray pump/X-ray probe setup we have captured the temporal evolution of X-ray induced changes in crystallized protein molecules, resulting among other things in an apparent increase of specific bond lengths. The magnitude of the increase depends on the pump probe delay. In time-resolved "molecular movie" experiments, changes in bond length of the order of 0.1 Å are attributed to

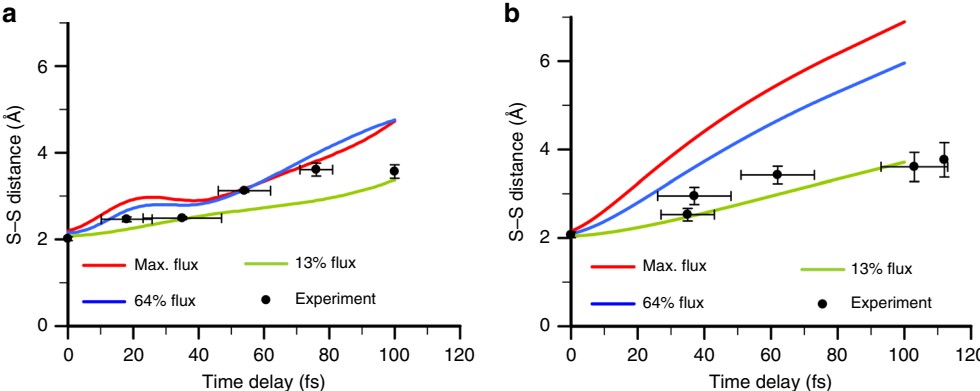

**Fig. 4 Theoretical analysis of sulfur displacements as a function of pump probe delay. a** Average distance between sulfur atoms of a S–S bridge as a function of pump-probe time delay in thaumatin, predicted by the XMDYN molecular dynamics simulation. The red, green and blue curves represent the S–S distances for the experimental total nominal fluence (pump and probe combined), $F_{max} = 7.0 \times 10^{12}$ photons $\mu m^{-2}$, $F_{med} = 4.4 \times 10^{12}$ photons $\mu m^{-2}$ (64% of $F_{max}$) and $F_{low} = 8.8 \times 10^{11}$ photons $\mu m^{-2}$ (13% of $F_{max}$), respectively, whereas the black curve with error bars represents the experimentally measured S–S separation with delay uncertainty and distance error included (see Supplementary Discussion). **b** Average distance between sulfur ions in a S–S bridge in lysozyme, as a function of pump-probe time delay, predicted by the hybrid continuum model. For direct comparison between the two models, the variations in the ionic charges in the hybrid model were matched to the corresponding XMDYN results. The good agreement between the two models confirms the importance of plasma electron screening and ion caging effects. The error bars of the experimental values correspond to the standard deviations of the XTCAV-derived time-delay values (*x*-axis) and give the standard deviation of the distribution (*y*-axis).

different chemical origins, which are then interpreted within a mechanistic framework. While our two-pulse experiment probes a somewhat different evolution of ionization dynamics than a single-pulse experiment, the time scales and effects are similar. Our experiments demonstrate an urgent need to revisit the assumptions of single-pulse SFX measurements (which often use 30–50 fs long pulses) and assess the degree to which such diffraction data is truly free of damage, including local damage to (metallo)cofactors and their ligands.

Our extensive experimental description serves as a test bed for theoretical analysis and method development. Two different theoretical methods obtained similar predictions on S–S separation confirming that the underlying mechanism has been correctly identified, highlighting the effects of ion caging and plasma ions. Currently, neither model incorporates a detailed description of bonding or chemical effects of the remaining bound electrons, which may be significant for understanding the unexpected dynamics of aromatic rings and the protein backbone. A quantitative description of the X-ray-induced dynamics in those chemical groups is beyond the reach of existing simulation tools and will require further theoretical development. Whether these are related to an ionization-related symmetry breaking of delocalized bonds remains to be seen. In contrast to fullerene[16] and xenon[17] crystals the coupling of the dynamics across unit cells is small in protein crystals due to screening effects.

The observation of XFEL-induced dynamics of functional groups embedded in a larger material has broader implications to non-crystalline biological systems, soft-matter phases and liquids, all of which can exhibit short-range atomic or molecular coordination. A deeper, more complete understanding of the molecular nature of local damage effects is clearly needed to inform the design of XFEL studies of such systems and ensure the correct interpretation of the results to obtain accurate chemical information. Our findings suggest future directions towards achieving this goal.

## Methods
**Nanocrystal preparation and characterization**. *Thaumatococcus daniellii* thaumatin was purchased from Sigma-Aldrich Chemie GmbH (Schnelldorf, Germany). Microcrystals were prepared by rapidly mixing 400 µl of 88 mg ml$^{-1}$ thaumatin in 100 mM Na Hepes pH 7.0, 400 µl of 1.6 M Na,K tartrate, and 20 µl of thaumatin

seed crystals. Microcrystals ($\sim 3 \times 3 \times 5 \mu m^3$) appeared overnight[22]. The microcrystalline slurry was centrifuged and washed twice with a buffer containing 0.1 M Na Hepes pH 7.0, 0.8 M Na,K tartrate. Microcrystals ($\leq 1 \times 1 \times 3 \mu m^3$) of hen egg white lysozyme (Sigma-Aldrich Chemie GmbH (Schnelldorf, Germany)) were grown by rapidly mixing cold solutions of protein (32 mg ml$^{-1}$ in 0.2 M Na acetate pH 3.0) and precipitant (20% NaCl, 6% PEG 6000, 1 M Na acetate pH 3.0) in a 1:3 ratio in a 4 °C cold room[22]. After overnight crystallization, the microcrystals were washed with storage solution (10% NaCl, 0.1 M Na acetate pH 4.0). To generate nanocrystals, microcrystals of lysozyme and thaumatin were filtered through 0.5 µm stainless steel filters using an HPLC system. This was followed by a 0.2 µm pore size manual filtration step for thaumatin. Prior to SFX data collection, the lysozyme nanocrystalline slurry was supplemented by 100 mM gadoteridol (Gd$^{3+}$:10-(2-hydroxypropyl)-1,4,7,10-tetraazacyclododecane-1,4,7-triacetic acid)[22].

**Data collection**. The experiment was performed in the nanofocus chamber of the Coherent X-ray Imaging (CXI) instrument[34] at the Linac Coherent Light Source (LCLS) in February 2015 (proposal LG07/LE70). The protein nanocrystals were introduced into the XFEL beam in a thin liquid jet using a gas dynamic virtual (GDVN) nozzle injector[35]. The position of the sample jet was continuously adjusted to maximize the hit rate. To follow the time-dependent X-ray–induced dynamics, an X-ray pump X-ray probe scheme was used[17] as shown in Supplementary Fig. 1a. Two 15-fs X-ray pulses were produced using the double-pulse operating mode at the LCLS[36], with a photon energy separation of ~80 eV centered on the iron K-edge at 7.112 keV. Our intent was to have very similar pulse energies for the X-ray pump and probe pulses and equally spaced time delays, increasing from 20 fs to 100 fs in 15–20 fs steps. However, due to very limited time for tuning of the accelerator neither goal was consistently met. Moreover, machine operation likely differed slightly from one accelerator operator to the next. This may explain inconsistencies in magnitude of the displacements observed in lysozyme.Gd for the nominally 20 fs and 40 fs time delays (determined by analyzing x-band transverse deflecting cavity (XTCAV) measurements[37] to be 35 and 37 fs, respectively). The experimentally derived values for the pulse energies (using gas detectors) and time delaysare listed in Supplementary Tables 1 and 2. No XTCAV values were recorded for the nominally 100 fs time delay for thaumatin and lysozyme.Gd. For the latter, it was reported to be 112 fs by the operators. We used the following experimental setup: The first X-ray pulse, with photon energy above the iron K-edge and pulse energy of ~0.5 mJ was used as a pump, inducing ionization dynamics in the system. The scattered X-rays were absorbed by an iron filter (thickness 25 µm) and did not reach the detector. The second X-ray pulse, with photon energy just below the iron K-edge and pulse energy of ~0.5 mJ, was used as a probe to measure Bragg diffraction, hitting the same sample segment. In this case, the scattered X-rays passed through the iron filter. Diffraction signals from the second X-ray pulse were recorded on a Cornell-SLAC Pixel Array Detector (CSPAD)[38] located ~70 mm downstream of the interaction region. We also collected a reference dataset using the probe pulse only by suppressing the pump probe upstream of the experimental chamber ("single pulse data"). At the beginning of each shift, the X-ray focus was optimized using imprints, a method by which the beam profile is deduced from the size of a vaporized area on a thin gold film hit by the beam at various intensity levels[39]. The data was collected in two shifts of 24 and 36 h, respectively. The diameter of the Gaussian X-ray focus was ~0.2 µm FWHM. With a beamline

transmission of ~45% the power density at the interaction region was nominally $2.7 \times 10^{19}$ W cm$^{-2}$ (corresponding to $3.5 \times 10^{12}$ photons μm$^{-2}$ per single pulse). The actual power density was likely lower[40].

**XTCAV analysis**. The pulse duration of the single pulses and the separation in time of the two pulses were measured using the XTCAV[41], located at the downstream end of the LCLS undulator. By measuring the longitudinal phase space of the electron bunch that created the XFEL photon pulse, one can obtain information about the laser power and temporal characteristics of each XFEL shot. However, information was stored only for each 4th XFEL pulse and not for all time-delays (see Supplementary Tables 1 and 2). Python scripts making use of the LCLS-provided *psana* analysis framework[42] were used to extract information on pulse length and pulse separation from the XTCAV data (Supplementary Fig. 1b). This information was then merged with the crystallographic data by means of the timestamp assigned to each individual XFEL shot. To set an uncertainty in time delay we used the standard deviation of the time delays present in each dataset.

**Data analysis and structure determination**. CASS[43,44] was employed for online data analysis, hit identification and data pre-processing. Indexing and integration were performed with CrystFEL[45] version 0.6.3. The positions and orientations of individual sensor modules of the CSPAD were refined as previously described[1]. The dose was calculated using RADDOSE 3D[26], without accounting for photoelectron escape. The single pulse data of lysozyme.Gd and thaumatin were phased by rigid-body refinement of PDB entries 4ETC[22] (lysozyme.Gd) and 5FGX[46] (thaumatin) after removal of water molecules and metal ions using REFMAC version 5.8.0222[47]. The final structures were obtained using iterative cycles of rebuilding in COOT[48] and refinement in REFMAC[47], resulting in a model with excellent geometry. Data and model statistics are given in Supplementary Tables 1 and 2. These structures were used as starting models for the refinement of the pump probe data using a jackknife approach (see below). PHENIX[49] was used to calculate the electron density maps.

**Refinement and jackknife analysis**. Refinement of the pump-probe data is complicated by the fact that the pump-induced ionization, increasing through the pulse, may not only modify the atomic form factors but also result in changes in nuclear positions, conflicting with standard atom and bond parameters. Neither effect is foreseen in refinement programs. In molecular refinement the target function to be minimized is commonly a weighted sum of a term that fits the X-ray data and a term that imposes geometric restraints. Thus care must be taken to prevent automatic "addressing" of any such changes by an increased B-factor; the relative weight between geometry and X-ray terms needs to be considered. We tested the influence of changing the weighting factor in REFMAC5[47] but ultimately decided on using the suggested value. Accordingly, changes in bond parameters are probably increasingly underestimated with longer pump probe delays: data quality decreases, shifting the weighting factor towards the geometry term. This affects, in particular, bonds involving low Z atoms (C,N,O). To avoid any influence of the normally employed refinement restraints on bond distances, angles and torsion angles, the disulfides were not modeled as normal S–S-bonded cysteines, but as two alanines with two nonbonded S atoms, each described as alternative conformations so as to even exclude van der Waals or other nonbonded interaction restraints. Otherwise, the possible elongation of the S–S bond is constrained and we observed difference density around the disulfides for the long time delay data, indicating insufficient separation. Changing the standard deviation for the S–S bond length had no influence. Refinement was considered converged when no difference electron density remained. For calculation of the isomorphous difference electron density maps, $F_{obs(\Delta t)} - F_{obs(single\ pulse)}$, the datasets were scaled using SCALEIT before computing the difference maps using RIDL[29]. To estimate the uncertainties in the atom-atom distances, we used a jackknife-type resampling method. Since the accuracy of the integrated intensities scales with the number of diffraction images, we used the maximal number of images available for each dataset (Supplementary Table 2). Starting from 100% of the images in the dataset, 100 datasets with 75% randomly chosen images were prepared. 100 structures were then refined against these datasets. The standard deviations observed for the atom-atom distances in these structures were then used as estimates for the errors in these values. Using all data should provide the best refinement models. However, since the data accuracy affects the refinement statistics and thus the error bars of the refined values, a comparison of absolute values determined for the different time delays becomes convoluted. Therefore, we repeated the jackknife analysis for the same number of images/dataset: using 11,000 images, 100 datasets were prepared by randomly choosing 9000 frames. These datasets were used to independently refine 100 models (Supplementary Table 1). The refinement results of the two approaches are essentially the same. The S–S bonds were superimposed in a common coordinate system to analyze the trajectories of the sulfur atoms (Supplementary Notes 1 and 2).

**Averaged difference electron density maps**. The maps used for local averaging were calculated after scaling to each other the structure factors obtained by Monte-Carlo averaging of 11,000 images for each time point, using SCALEIT including Wilson scaling. For all difference maps, a high resolution limit of 2.3 Å was used.

Phases were obtained from structures refined against the single-pulse datasets. Local averaging of difference density maps was performed with custom-written python scripts available from https://github.com/tbarends/LocalAveraging. These scripts calculate the difference electron density on a predefined grid set up around each peptide moiety or each sidechain of either phenylalanine, tyrosine or tryptophan. They then calculate the average of these maps, which is then written out as an XPLOR format map file for visualization.

**Bragg termination simulation**. We simulated the effect of Bragg termination[50] on the Wilson and intensity distribution statistics by modifying the Bragg intensities of the single pulse data according to Barty et al.[50]. Since we cannot simulate the effect of the "evolution" time between pump and probe pulse on the sample, we used a probe pulse length of 100 fs. We assumed an increase of the average atomic displacement $B_{eff}$ with the third power of time from 0 Å$^2$ at the beginning of the 100 fs long pulse to 125 Å$^2$ and 500 Å$^2$ for thaumatin and lysozyme.Gd, respectively, at the end of the pulse. An effective time-dependent B-factor $B_{eff}(t)$ was defined for each step and increased with the third power of time. For each time point $t_p$, $B_{eff}(t_p)$ was applied to the single pulse intensities. The resulting new intensity at time $t_p$ were averaged over all time points to obtain the Bragg termination-simulated intensities.

**XMDYN simulations**. For the simulation, we used XMDYN, a molecular-dynamics-based and Monte-Carlo-based code for modeling X-ray driven dynamics in complex systems[30–32]. It is coupled on-the-fly to the atomic structure calculation tool, XATOM[30,51,52]. This many-body and fully non-equilibrium model takes into account all relevant X-ray induced processes in matter (such as atomic photoionization, inner-shell Auger and fluorescent decay, collisional ionization and recombination), using their microscopic description. Chemical bonds can be represented using classical force fields[31,53]. However, in the current study, they were not included. XMDYN simulations follow the temporal evolution of a stochastically ionized system. The code has been successfully applied to describe the interaction of clusters and macromolecules with X-rays[31,32,53,54]. It is not computationally feasible to simulate the dynamics of a non-uniformly irradiated crystal with a size of a few hundred nm using a code that follows the trajectories of all atoms and electrons individually. Instead, we used the following approximation: We calculated the dynamics of the atoms and electrons within a smaller cubic volume of nm size (here called 'supercell') exposed to a pulse of a spatially uniform fluence and imposed periodic boundary conditions[55,56]. Ideally, one would choose the crystallographic unit cell of thaumatin as the supercell. However, such calculations would still be too demanding, due to the large number of the particles involved. Therefore, as a supercell, we used a reduced, $S_x = S_y = S_z = 14$ Å size subunit containing ~300 atoms, including one disulfide bridge. We 'cut out' the subunit from the solvated crystallographic unit cell of thaumatin (pdb: 1RQW) in such a way that the ratio of sulfur density to the density of light atoms was the same as in the full crystallographic unit cell of thaumatin. By performing the simulation at various fluence values, we investigated the ionization dynamics occurring in various regions of the crystal with respect to the beam focus. We also performed a convergence study by increasing the size of the supercell. The characteristics of the particle dynamics showed no significant changes with respect to the 14 Å case, ensuring that the extracted results reliably represent the response of a large system.

We considered pulse parameters based on the nominal values reported from the experiment. We set the photon energy for both pump and probe pulses to 7.14 keV, neglecting the small difference in the photon energy of the two pulses (as it is not relevant to the dynamics). We used a Gaussian temporal pulse profile with 15 fs FWHM. Three different total fluence values (pump and probe combined, each with a half of the total fluence) were considered in the simulations: (i) high fluence of $F_{max} = 7.0 \times 10^{12}$ ph μm$^{-2}$, expected at the center of the beam focus, (ii) medium fluence of $F_{med} = 4.4 \times 10^{12}$ ph μm$^{-2}$ (64% of maximum fluence), and (iii) low fluence of $F_{low} = 8.8 \times 10^{11}$ ph μm$^{-2}$ (13% of maximum fluence). Different time delays were considered between 0 fs and 100 fs. For the analysis, 100 XMDYN trajectories were calculated for the three fluence cases, in order to provide sufficient statistics for the data analysis, see Supplementary Note 3.

**Hybrid plasma/molecular dynamics simulation**. The hybrid plasma-MD model consists of three stages: (i) atomic transition rate calculations, (ii) a rate-equations calculation to estimate mean plasma parameters, and (iii) ion dynamics calculation. The photoionization, Auger and fluorescence rates are calculated using a non-relativistic quantum code[57]. These rates are sufficiently similar to those obtained by XATOM that differences in the values are not expected to significantly affect the results of the simulations.

Electron parameters for the plasma model were calculated using rate equations[58]. The computation uses the ionization and decay rates obtained as described above and secondary impact ionization rates from the literature (see references in ref. [59]). Ejected electrons are assumed to be trapped if their kinetic energy exceeds the trapping energy of the ionized molecule. We assume a spherical geometry to determine if ejected electrons are trapped. This is the only place where geometry is included in the rate equations calculation. Both photoelectrons and some of the Auger electrons have sufficient energy to escape at early times. The calculation assumed instantaneous thermalization of electrons ejected in the

ionization process apart from photoelectrons, which are only trapped if the attractive potential of the positively particle is sufficiently strong[58]. The thermalization assumption leads to higher ion charge states when compared to the XMDYN simulation (see Supplementary Fig. 15). The key physical parameters obtained from the rate equations calculation are the ion charge states, and the density and temperature of the trapped electron gas.

For the ion dynamics simulation with continuum treatment of the trapped electrons the model assumes pairwise interactions of ions, labeled $i$ and $j$, separated by a distance $r_{ij}$. The effective interaction pair potential is taken to be a screened Debye interaction, $V_D(r_{ij}, t)$, plus a residual interaction that may be configured either as a short-range collision potential, $V_C(r_{ij}, t)$, or another pairwise interaction $V_B(r_{ij}, t)$.

The interaction of the crystalline target with the pump XFEL pulse liberates electrons that form plasma, modifying the electrostatic interactions between ions. The screened electrostatic interaction is of the conventional Debye form

$$V_D(r_{ij}, t) = \frac{Q_i(t)Q_j(t)}{r_{ij}} \exp[-\lambda_D(t)r_{ij}] \quad (1)$$

where $Q_i(t)$ is the ionic charge of ion $i$ as determined by quantum mechanical calculations of the response of its atomic species to the electromagnetic parameters of the incident XFEL radiation. The parameter $\lambda_D(t)$ is the Debye length, which is calculated from the temperature and density of the electron plasma present at time $t$ during the interaction.

An ionic collision potential may be employed to exclude an unphysically close approach of two ions:

$$V_C(r_{ij}, t) = D_{ij}(t)[1 - \exp(-a_{ij}(t)(r_{ij} - r_{ij}^e(t)))]^2 \text{ for } 0 \leq r_{ij} \leq r_{ij}^e(t)$$
$$= 0 \text{ otherwise} \quad (2)$$

The parameters $r_{ij}^e(t)$ are chosen based on the effective radii of the interacting ionic species. The remaining parameters may be adjusted to modify the "hardness" of the collisional interaction, which takes effect whenever any two ions are separated by a critical length $r_{ij}^e$.

The values of the parameters $Q_i(t)$ and $\lambda_D(t)$ are determined by the rate equations model. They depend on the atomic species involved, the incident intensity of the X-ray radiation and the presumed physical size of the irradiated crystals. The parameters of the Debye potential are derived from estimates of the density and temperature of the plasma and are utilized as input parameters to the molecular dynamics simulation. These simulations track the motions of ions under the influence of the forces that act according to

$$\vec{F_i} = -\sum_j \vec{\nabla} V_{ij} \quad (3)$$

The trajectories of the ions are determined by solving coupled equations of motion using standard Runge-Kutta integration, on the assumption that the ions are at rest in their equilibrium positions at time $t = 0$. A spherical simulation target region is defined, centered on the center of the S–S bond. All atoms within this region are free to move, while all other atoms in the unit cell are fixed. The electrostatic interactions that act on the target atoms are calculated by summing periodic replications of the unit cell, though the steady decrease of the value $\lambda_D(t)$ towards atomic length scales ultimately restricts that summation to just a few tens of Ångstrom. This greatly reduces the time required to complete the simulation. In hybrid plasma/MD model the size of the crystal determines the photo-electron escape rate. For experimental values of the incident fluence the contribution to the ionization dynamics by photo-electrons is small compared to secondary electrons that are produced shortly after the start of the pulse and are mostly trapped. A change in the crystal size weakly affects the predicted ion charges and resulting atomic motion between the pulses; these changes were captured by simulations over a distribution of particle sizes. Also, the changes in the rates of underlying atomic electronic processes on the scale of 10–30% translate into 2–5% changes in predicted charges[57] which has negligible effect on the observed motion of atoms.

**Reporting summary**. Further information on research design is available in the Nature Research Reporting Summary linked to this article.

## Data availability
Coordinates have been deposited with the PDB (Accession codes: 6SRJ, 6SRQ, 6SRK, 6SRL, 6SRO, 6SRP, 6SR0, 6SR1, 6SR2, 6SR3, 6SR4, 6SR5). The source data underlying Figs. 1b–d, 2e, 4a, b and Supplementary Figs. 8a, b and 15a–c are provided as a Source Data file. Other data are available from the corresponding authors upon reasonable request.

## Code availability
Analysis scripts are available from the corresponding authors upon request.

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

## Acknowledgements
This research was supported by the Max Planck Society. The work performed by the Hamburg group was supported in part by the Cluster of Excellence 'Advanced Imaging of Matter' of the Deutsche Forschungsgemeinschaft (DFG)-EXC 2056-project ID 390715994. We thank Dr. Roland van Gessel, Bracco Imaging Deutschland, Konstanz, Germany, for the very generous gift of the sample of gadoteridol and Beatrice Latz for support making capillaries. Use of the Linac Coherent Light Source (LCLS), SLAC National Accelerator Laboratory, is supported by the U.S. Department of Energy, Office of Science, Office of Basic Energy Sciences under Contract No. DE-AC02-76SF00515. Parts of the sample injector used at LCLS for this research was funded by the National Institutes of Health, P41GM103393, formerly P41RR001209.

## Author contributions
S.B. conceived the experiment which was designed and coordinated by S.B. and I.S.; G.N., E.H., I.S., and R.L.S. prepared samples; A.A.L., F.J.D., and A.M. lead the setup and delivery of the two-color FEL setup; R.B.D., G.N.K., and R.L.S. performed sample injection, S.B., M.S.H., A.A., and M.H. operated CXI and collected data; C.M.R., M.H., K.N., and L.F. performed online processing; K.N., L.F., A.G. performed offline analysis; K.N., A.G., and T.R.M.B. analyzed diffraction data; M.K., K.N. refined structures; A.K., A.V.M., and H.M.Q. performed hybrid model calculations; M.M.A., Z.J., R.S., B.Z. developed the XMDYN-based modeling strategy. M.M.A. performed the XMDYN calculations. S.B., L.F. contributed discussions. T.R.M.B. supervised crystallographic analysis, B.Z., H.Q., R.S. supervised calculations. All authors discussed the results and contributed to the manuscript. The initial versions were written by I.S., A.V.M. and B.Z.

## Competing interests
The authors declare no competing interests.
