## [Peer Review File · Nature Communications]

Reviewers' comments:

Reviewer #1 (Remarks to the Author):

Review of Nass et al

This is an important, careful and thorough study of great impact for the use of Free Electron Lasers to obtain 'damage free' macromolecular structures, since it reports an analysis of damage as a function of time delay. It shows that it is essential to pay attention to the XFEL pulse length if 'radiation damage free' structures are to be obtained. The data are complemented by detailed theoretical calculations and simulations which enhance the study and are validated by the XFEL experimental observations (or vice versa depending on particular viewpoints).

The only major omission from the manuscript which the authors could choose to address is to comment more extensively on the implications of their results for general macromolecular crystallography data collection strategies at XFELs (i.e. expand on their statement in the conclusions that there is an 'urgent need to revisit the assumptions of single-pulse SFX measurements...').

In terms of readability, a minor point is that the paper overuses the first person plural ('we' and 'our') and would benefit from a number of these being changed to the passive voice.

Overall, the manuscript will have significant influence on the field and is highly appropriate for publication in Nature Communications following some minor revision and edits as detailed below.

Main paper minor comments:

P2.

Abstract. 'The need for very short XFEL pulses can be... damage progresses.' This sentence could be rewritten to explain the technical phrase 'gating of Bragg diffraction' in order not to perplex readers so soon into the paper.

Line 3. Change 'to capture' to 'the capture of'

P3.

Line -10. Change 'studying' to 'the study of'

Line -8. Insert beam size into this paragraph as this is an important parameter relative to the crystal size.

Line -2. 'single-pulse data'. First mention of the single-pulse experiment so is confusing. Needs some explanation, or a sentence adding at the bottom of P3 in the Introduction that this additional experiment was carried out.

P4.

Paragraph 2. Were the doses calculated with or without the inclusion of photoelectron escape from the crystals? State which.

P5.

Line 5. Perhaps add 'and so increasing the dose' after 'highly ionized atoms'.

Line 8. Add ',' after 'maps'

Line -15 'trajectories of sulphur atoms'. Might be a good place to mention the rough velocity the authors have calculated for the sulphurs (1000 m/s) that at the moment is only stated in the Suppl Mat.

P6.

Line 9. Add ',' after 'irradiation'

Line 12. Add 'cases of' before 'modelled' and delete 'cases after 'fluence'

Line 13. Add 'for the calculations' after 'In particular,' to remind the reader that this section is all about simulations.

Line -14. For better clarity, change 'Increasing the fluence' to 'Increasing the fluence in vacuum' (if this is what you mean).

Line -12. 'This proves' is very/too strong. 'This shows' would be much better.

P8.

Add size of beam to data collection section. This is a pivotal parameter. For the veracity of the

results.

P9.

Line 8. Add 'experimentally' after 'determined' and change 'The experimentally' to 'These experimentally'

Line 10. Change 'Supplementary Table 1 to 'Supplementary Tables 1a and b'

Line 11 'using imprints'. What are these? Explain.

Line -1. 'Supplementary Table 1 to 'Supplementary Tables 1a and b'

P10

Line 9. 'RADDPOSE_3D' with or without inclusion of photoelectron escape?

Line 15. 'jackknife approach' Add '(see below)' here. This will be a new and unknown approach to many readers.

Line -12. Change 'x-ray' to 'X-ray' to be consistent.

The refinement and jackknife analysis reported for this study has been well thought out and executed. It covers a number of traps including model bias and scaling artefacts.

P11

Line -16. 'of the two approaches are the same'. Were they really identical (which is what saying 'the same' would imply) or just 'very similar'?

P12

4 more uses on 'we'

P13

Line -11. Insert '.' After 'Ref 58)'

P14

Line -4. 'The values of....'being irradiated.' Would be useful to include some idea of the sensitivity of the results to changes in the input parameters e.g. crystal size: by how much does a small change in the presumed size alter the results?

P15

Line 8. Add a comment on the limitations imposed by restricting the summation to a few tens of Angstroms: what are the issues with doing this?

Supplementary Material minor comments:

P4. Line 6. This is the first mention of the velocity of the sulphur atoms and is an interesting number, even though approximate. Suggest including it in the main paper somewhere.

P5. Line -10. 'We conjecture that the conformation of the disulphide bridges....' Indeed, a correlation between disulphide bridge conformation and susceptibility to radiation damage has been noted by Gerstel et al J. Synchrotron Rad. (2015). 22, Fig 8.

P7. Line -6. 'and is even higher for sulphur' would read better than 'for sulphur even higher'
2.1.2 Here is a good example of the overuse of 'We': 3 out of 4 paragraphs start with it and the 4th paragraph has it as the second word.

P8. Line 7. Better to say 'shown in Supplementary' rather than 'from Supplementary'

P8. Line-6. Better to replace 'charging' by 'increasingly charged'

P11. Re comments in the first paragraph on the spatial and temporal distribution of photons in the XFEL beam, the authors might also comment on the variable intensity profile of the beam in terms of its energy, since that is another source of uncertainty in the experiments.

P18. Line 2. Change 'rather normal' to 'as expected' ('normal' has a special meaning re distributions and so it would be better not to use it here.

P19. Line 4. Fix 'Specifically, ,'

P19. Line 8 Change 'in B with' to 'in B-factor with'

P20. Line 1. Change '160 were' to '160 steps were'

P22. Suppl Figure 8. a and b require right hand y-axes for the Pulse energy (mJ).

P23. Line 3. Add 'Pm' after "midpoint" and in line 4 add 'Pi' after "intersection point"

P27. Suppl Figure 11. State what radii of atoms have been used in these figures – are they van der Waals radii?

P31. last line. 'shadowing of the detector' shadowed by what? Clarify this statement.

P35. Add '(7 × 10¹² ph/m²)' after 'maximum fluence case'

P37-42. Tables 1a and b. Add Å units to all wavelength rows.

Elsbeth F Garman

Reviewer #2 (Remarks to the Author):

XFEL is a new light source, and the nature of its interaction with matter remains to be elucidated. The authors have been studying the radiation damage of protein samples by XFEL irradiation since the launch of LCLS. The author's experimental design in this paper is novel and elaborate. The processing of the data obtained is precise and the results are new and interesting. The discussion seems logical and reasonable.

Although simulations are only predictions, the authors' challenges should be appreciated because they provide hints and clues for understanding the phenomenon of radiation damage. So, I totally agree with the author's view "Our extensive experimental description serves as a test bed for theoretical analysis and method development".

The interesting results in this paper will be welcomed not only in the fields of XFEL crystallography and structural biology, but also in the fields of physics and chemistry, and will have a significant impact on the progress of science. Therefore, I consider this paper worthy of being published in Nature Communications.

I would like to ask the authors for the following questions. If possible, I would suggest incorporating these points into the manuscript.

Q1:

In Figure 2 (c and d), positive densities disappeared as the delay time increased. On the other hand, In Figure 3 (c), negative densities disappeared as the delay time increased. Are these phenomena due to the actual recovery from radiation damage? What is the expected mechanism? Is the interaction of hydrated electrons involved?

Q2:

In Conclusion section, the authors described "Our experiments demonstrate an urgent need to revisit the assumptions of single-pulse SFX measurements (which often use 30-50 fs long pulses) and assess the degree to which such diffraction data is truly free of damage, including local damage to (metallo)cofactors and their ligands."

Compared to LCLS, SACLA uses shorter duration pulses: 2–10 fs (Fukuda et al & Mizohata, PNAS, 2016). What do you think of radiation damage in cases using shorter pulses such as SACLA? Can it be predicted by simulations?

Reviewer #3 (Remarks to the Author):

The authors performed XFEL experiments and computational analyses to examine the effect of strong XFEL pulse on the atomic positions. This is an important study that may provide some answers to the concerns regarding serial-femtosecond crystallography studies – whether very strong XFEL pulses are affecting the protein structures.

Overall, the results are convincing, but some parts of the texts were difficult to follow; there are quite many discussions referring to Supplementary Information. It may be due to space limitations, but it would be easier to read if the data are described more in the main text. For

example, in page 3, bottom part, "intensity of ... in the focus, see Supplementary Discussion"); it could help if we can know what's in the Supplementary Discussion.

Some specific comments:

1. Page 4, the last sentence: "The statistics of the single-pulse data are worse than that of the double-pulse data, most likely due to the significantly lower pulse energy". What "statics" are worse? The energy of the single-pulse data is half of the double-pulse data if I understand correctly. Does it become so bad?
2. "Theoretical analysis" focuses on S-S bond. But the effects on protein residues are also very important. Why was it done just for S-S bond?
3. "continuum electron plasma model". It's not obvious why two theoretical models were considered. What is insufficient in XMDYN analysis? What does "plasma model" provides?
4. page 7. "SFX measurements (which often use 30-50 fs long pulses)." Some studies were done using shorter pulses, <10 fs at SACLA.
5. page 10. "The restraints for S-S bonds, torsion angles etc. were modified to contain two non-bonded S atoms and two alanine residues, described as alternative conformations to exclude van der Waals interactions". This sentence is not clear. This treatment may have important effects on the model refinement and should be described in detail.

Figure 4. According to this comparison of calculations and experimental data, only 13% of Max Fluence best matches the data. Is it expected?

We thank all reviewers for their comments that have improved our manuscript.

Reviewer #1 (Remarks to the Author):

Review of Nass et al

This is an important, careful and thorough study of great impact for the use of Free Electron Lasers to obtain 'damage free' macromolecular structures, since it reports an analysis of damage as a function of time delay. It shows that it is essential to pay attention to the XFEL pulse length if 'radiation damage free' structures are to be obtained. The data are complemented by detailed theoretical calculations and simulations which enhance the study and are validated by the XFEL experimental observations (or vice versa depending on particular viewpoints).

The only major omission from the manuscript which the authors could choose to address is to comment more extensively on the implications of their results for general macromolecular crystallography data collection strategies at XFELs (i.e. expand on their statement in the conclusions that there is an 'urgent need to revisit the assumptions of single-pulse SFX measurements...').

In terms of readability, a minor point is that the paper overuses the first person plural ('we' and 'our') and would benefit from a number of these being changed to the passive voice.

We are very grateful for both the scientific comments/questions and the careful editing of our manuscript. Both improve it. We followed the recommendations as outlined below and also reduced the use of "we".

Overall, the manuscript will have significant influence on the field and is highly appropriate for publication in Nature Communications following some minor revision and edits as detailed below.

Main paper minor comments:

P2.

Abstract. 'The need for very short XFEL pulses can be... damage progresses.' This sentence could be rewritten to explain the technical phrase 'gating of Bragg diffraction' in order not to perplex readers so soon into the paper.

Line 3. Change 'to capture' to 'the capture of' -----done

P3.

Line -10. Change 'studying' to 'the study of' -----done

Line -8. Insert beam size into this paragraph as this is an important parameter relative to the crystal size.

-----done

Line -2. 'single-pulse data'. First mention of the single-pulse experiment so is confusing. Needs some explanation, or a sentence adding at the bottom of P3 in the Introduction that this additional experiment was carried out. -----*We agree and added a sentence.*

P4.

Paragraph 2. Were the doses calculated with or without the inclusion of photoelectron escape from the crystals? State which. ----- *Without. We state this now*

P5.

Line 5. Perhaps add 'and so increasing the dose' after 'highly ionized atoms'.

---- *this now reads "increasing the concentration of highly ionized atoms and thus dose"*

Line 8. Add ',' after 'maps' -----done

Line -15 'trajectories of sulphur atoms' . Might be a good place to mention the rough velocity the authors have calculated for the sulphurs (1000 m/s) that at the moment is only stated in the Suppl Mat.

P6. -----*we added it here: "The overall magnitude, speed (~ 1000 m/s) and kinetics of the average S-S bond elongation are very similar for both lysozyme.Gd and thaumatin despite the very different structures and dose (Fig. 1)."*

Line 9. Add ',' after 'irradiation' -----done

Line 12. Add 'cases of' before 'modelled' and delete 'cases after 'fluence' -----done

Line 13. Add 'for the calculations' after 'In particular,' to remind the reader that this section is all about simulations. -----done

Line -14. For better clarity, change 'Increasing the fluence' to 'Increasing the fluence in vacuum' (if this is what you mean). ---- *we agree, done*

Line -12. 'This proves' is very/too strong. 'This shows' would be much better. -----done

P8.

Add size of beam to data collection section. This is a pivotal parameter. For the veracity of the results.

P9. -----*You probably missed this; it is given towards the end of the experimental section "The diameter of the Gaussian X-ray focus was ~0.2 μm FWHM."*

Line 8. Add 'experimentally' after 'determined' and change 'The experimentally' to 'These experimentally' ----- *We modified this part.*

Line 10. Change 'Supplementary Table 1 to 'Supplementary Tables 1a and b' -----done

Line 11 'using imprints'. What are these? Explain. ---- *We added the following sentence to the manuscript: "At the beginning of each shift, the X-ray focus was optimized using imprints, a method by which the beam profile is deduced from the size of a vaporized area on a thin gold film hit by the beam at various intensity levels. "*

We are citing <https://journals.aps.org/prapplied/abstract/10.1103/PhysRevApplied.4.014004> This publication describes a significantly more detailed analysis than we did. It however has all the relevant earlier references cited in it.

Line -1. 'Supplementary Table 1 to 'Supplementary Tables 1a and b' -----done

P10

Line 9. 'RADDOSE_3D' with or without inclusion of photoelectron escape? ---without (now added)

Line 15. 'jackknife approach' Add '(see below)' here. This will be a new and unknown approach to many readers. -----done

Line -12. Change 'x-ray' to 'X-ray' to be consistent. -----done

The refinement and jackknife analysis reported for this study has been well thought out and executed. It covers a number of traps including model bias and scaling artefacts. ---- Thank you very much, we agree.

P11

Line -16. 'of the two approaches are the same'. Were they really identical (which is what saying 'the same' would imply) or just 'very similar'? ---- they were the same within error.

P12

4 more uses on 'we'

P13

Line -11. Insert '.' After 'Ref 58)' -----done

P14

Line -4. 'The values of...'being irradiated.' Would be useful to include some idea of the sensitivity of the results to changes in the input parameters e.g. crystal size: by how much does a small change in the presumed size alter the results?

In the hybrid plasma/MD model the variation in the finite size of the crystals affects the photoelectron escape rate. Secondary electrons dominate the ionization dynamics shortly after the start of the pulse for experimentally relevant values of the incident fluence. Secondary electron temperature and density depend weakly on the crystal size as they are almost always trapped. Hence, the theoretical prediction for atomic displacement is not significantly affected by the size of the crystal. Also, reference 51 (now 57) (A Kozlov and H M Quiney Phys. Scr. 94 075404 (2019)) discusses the variations in the solutions of the rate equations when the underlying atomic rates are obtained from different electronic structure models (Hartree-Fock or local density) and differ by up to 10 – 30 %. It was shown that the solutions of the rate equations under the typical values of XFEL fluence vary by no more than 2 – 5% for light chemical elements present in the samples discussed in this work.

We added this to the end of the Methods section.

P15

Line 8. Add a comment on the limitations imposed by restricting the summation to a few tens of Angstroms: what are the issues with doing this?

The number of atoms confined to a spherical shell of radius R is proportional to R^2 . If the interaction in question is long range (Coulomb, $\sim R^{-2}$), the force at which atoms on the spherical shell act on an atom at its centre can be estimated to vary as $R^{-2}R^2 \sim O(1)$ which is not negligible. However, the Debye screening results in a short range interaction that is exponentially suppressed by a factor $\exp\left(\frac{-R}{\lambda}\right)$ where λ is the Debye length.

We added this to the Supplementary Information after “If the asymptotic...”

Supplementary Material minor comments:

P4. Line 6. This is the first mention of the velocity of the sulphur atoms and is an interesting number, even though approximate. Suggest including it in the main paper somewhere. -----
done

P5. Line -10. ‘We conjecture that the conformation of the disulphide bridges....’ Indeed, a correlation between disulphide bridge conformation and susceptibility to radiation damage has been noted by Gerstel et al J. Synchrotron Rad. (2015). 22, Fig 8. ----- *Thank you, we added the reference*

P7. Line -6. ‘and is even higher for sulphur’ would read better than ‘for sulphur even higher’ ----
---done

2.1.2 Here is a good example of the overuse of ‘We’: 3 out of 4 paragraphs start with it and the 4th paragraph has it as the second word. ---- *addressed*

P8. Line 7. Better to say ‘shown in Supplementary’ rather than ‘from Supplementary’ -----
done

P8. Line-6. Better to replace ‘charging’ by ‘increasingly charged’ -----*done*

P11. Re comments in the first paragraph on the spatial and temporal distribution of photons in the XFEL beam, the authors might also comment on the variable intensity profile of the beam in terms of its energy, since that is another source of uncertainty in the experiments.----- *This is correct, thank you. We changed the sentence to “The spatial and temporal distribution of the photons in an XFEL pulse, as well as the pulse energy, can change from shot to shot. While the latter is measured and known on every pulse, there is no shot to shot measurement of the spatial profile. Some limited information on the temporal profile of the beam can be extracted from the XTCAV measurement”*

P18. Line 2. Change ‘rather normal’ to ‘as expected’ (‘normal’ has a special meaning re distributions and so it would be better not to use it here. -----*done*

P19. Line 4. Fix ‘Specifically, ,’ -----*done*

P19. Line 8 Change ‘in B with’ to ‘in B-factor with’ -----*done*

P20. Line 1. Change ‘160 were’ to ‘160 steps were’-----*done*

P22. Suppl Figure 8. a and b require right hand y-axes for the Pulse energy (mJ). -----*done*

P23. Line 3. Add ‘Pm’ after “midpoint” and in line 4 add ‘Pi’ after “intersection point” -----
done

P27. Suppl Figure 11. State what radii of atoms have been used in these figures – are they van der Waals radii? -----*We used 1.8 Å radius for all atoms and added this information to the figure legend.*

P31. last line. 'shadowing of the detector' shadowed by what? Clarify this statement. We added *"by the injector shroud"*

P35. Add '(7 × 10¹² ph/m²)' after 'maximum fluence case' -----*done*

P37-42. Tables 1a and b. Add Å units to all wavelength rows. -----*done*

Elspeth F Garman

Reviewer #2 (Remarks to the Author):

XFEL is a new light source, and the nature of its interaction with matter remains to be elucidated. The authors have been studying the radiation damage of protein samples by XFEL irradiation since the launch of LCLS. The author's experimental design in this paper is novel and elaborate. The processing of the data obtained is precise and the results are new and interesting. The discussion seems logical and reasonable.

Although simulations are only predictions, the authors' challenges should be appreciated because they provide hints and clues for understanding the phenomenon of radiation damage. So, I totally agree with the author's view "Our extensive experimental description serves as a test bed for theoretical analysis and method development".

The interesting results in this paper will be welcomed not only in the fields of XFEL crystallography and structural biology, but also in the fields of physics and chemistry, and will have a significant impact on the progress of science. Therefore, I consider this paper worthy of being published in Nature Communications.

I would like to ask the authors for the following questions. If possible, I would suggest incorporating these points into the manuscript.

Q1:

In Figure 2 (c and d), positive densities disappeared as the delay time increased. On the other hand, In Figure 3 (c), negative densities disappeared as the delay time increased. Are these phenomena due to the actual recovery from radiation damage? What is the expected mechanism? Is the interaction of hydrated electrons involved?

The order of the crystalline lattice decreases with the time delay between the X-ray pump and probe pulses, respectively, due to uncorrelated motion of the randomly ionized atoms. This

means that the intensities of high resolution Bragg reflections decrease and ultimately disappear. Consequently it becomes harder and harder to distinguish small differences between the single pulse diffraction intensities and the increasingly noisy diffraction intensities collected at longer time delays. We believe this is the main reason why the magnitude of the difference densities decrease with delay time in both Fig. 2c,d and Fig. 3c. The underlying changes in Fig. 2c,d (backbone atom movement) are larger than the changes in the aromatic rings (which we cannot explain with a structural model), therefore they are visible for longer time delays. We do not know to which extent hydrated electrons are involved.

[Both the XMDYN and hybrid plasma/MD models employed point-like treatment of atoms which ignores molecular effects in the description of the electron density evolution in molecules. Electron recombination, which was included in the theoretical models, is insufficient to explain the observed electron density recovery, especially in light element complexes.]

Q2:

In Conclusion section, the authors described "Our experiments demonstrate an urgent need to revisit the assumptions of single-pulse SFX measurements (which often use 30-50 fs long pulses) and assess the degree to which such diffraction data is truly free of damage, including local damage to (metallo)cofactors and their ligands."

Compared to LCLS, SACLA uses shorter duration pulses: 2–10 fs (Fukuda et al & Mizohata, PNAS, 2016). What do you think of radiation damage in cases using shorter pulses such as SACLA?

It is enticing to speculate that SACLA data, or in general data collected with shorter pulse durations are less affected by radiation damage. However, in the absence of data demonstrating this we would prefer to refrain from such speculations.

Can it be predicted by simulations?

For equal numbers of photons in two pulses of different duration, a sub 10 fs pulse leads to a smaller ionization []. Auger processes have timescales in light elements of 10.7 fs (C), 7.1 fs (N), and 4.9 fs (O). This implies that upon exposure with pulse of a duration below 10 fs, fewer Auger electrons will be emitted during the pulse. As Auger electrons and their secondaries are the main contributors to the electronic radiation damage, one can achieve a suppression of the total radiation damage by reducing Auger electron emission [*,51]. Also, a sub 10 fs pulse leaves less time for secondary ionization to develop. Although the charges on individual atoms build up, there is less time of exposure left for atoms to move (one can estimate the magnitude of atomic motion to be at least linear in pulse length) under resulting Coulomb forces. However, the electron density evolution may be more visible for shorter pulses as the lack of atomic motion suppresses the damage gating effect for higher fluences. We also expect simulations to be more*

accurate for shorter pulses as secondary damage mechanisms that are sensitive to molecular effects play a relatively smaller role.

[] Ziaja, B., Jurek, Z., Medvedev, N., Saxena, V., Son, S.K. & Santra, R., Towards Realistic Simulations of Macromolecules Irradiated under the Conditions of Coherent Diffraction Imaging with an X-ray Free-Electron Laser. *Photonics* **2**, 256-269 (2015).*

*[51] Son, S. K., Young, I. D. & Santra, R. Impact of hollow-atom formation on coherent x-ray scattering at high intensity. *Phys. Rev. A* **83**, 069906 (2011).*

Reviewer #3 (Remarks to the Author):

The authors performed XFEL experiments and computational analyses to examine the effect of strong XFEL pulse on the atomic positions. This is an important study that may provide some answers to the concerns regarding serial-femtosecond crystallography studies – whether very strong XFEL pulses are affecting the protein structures.

Overall, the results are convincing, but some parts of the texts were difficult to follow; there are quite many discussions referring to Supplementary Information. It may be due to space limitations, but it would be easier to read if the data are described more in the main text. For example, in page 3, bottom part, “intensity of ... in the focus, see Supplementary Discussion”); it could help if we can know what’s in the Supplementary Discussion.

Some specific comments:

1. Page 4, the last sentence: “The statistics of the single-pulse data are worse than that of the double-pulse data, most likely due to the significantly lower pulse energy”. What “statics” are worse?

We thank the referee for his/her question; it made us realize that this statement is not true as written. The single-pulse statistics are worse than the nominal 20 fs time delay stats, but better than the nominal 100 fs statistics. We have therefore modified the sentence to “Statistics such as R_{split} , CC and $1/\sigma$ of the single-pulse data are worse than that of the double-pulse data at short time delays”.

The energy of the single-pulse data is half of the double-pulse data if I understand correctly. Does it become so bad?

This is correct. This means that the weak high resolution reflections have much lower signal:noise ratio. In addition to being collected with weaker X-ray pulses (lower pulse energy) the single pulse data of lysozyme suffer more from the shadowing of the detector from the detector shroud. We only realized this later.

2. “Theoretical analysis” focuses on S-S bond. But the effects on protein residues are also very important. Why was it done just for S-S bond?

The dynamical behaviours of light elements (H, C, N, O) and heavy (P, S, metals) are quite different because the latter more rapidly achieve higher charge states under the experimental conditions considered here. An underlying assumption of both of the computational models that we have considered is that the nuclear positions may be determined by considering the motions of point ions. This is a reasonable assumption for the sulphur ions, which occur in bridged pairs; the accumulation of charge causes rapid acceleration of the ions in the pair through mutual repulsion that is the most significant event during the early history of the interaction. The sulphur ions subsequently collide with other parts of the structure. The lighter elements, on the other hand, do not achieve such high charge states and it is likely that they remain bonded or dissociate into bonded fragments. The observed behaviour of the protein residues suggests behaviour that is beyond the scope of the simple ionic model adopted in our simulations.

3. “continuum electron plasma model”. It’s not obvious why two theoretical models were considered.

As we write in the manuscript on P. 6, the idea to consider two complementary theory models was to strengthen the manuscript's conclusions by ensuring that they are not biased by a particular model implementation or deficiency (particularly, the different representations of free electrons in the two models). Both models, XMDYN and the plasma model, were thoroughly and consistently constructed within the defined approximation framework and then carefully tested. The analysis with both models consistently shows that the experimentally observed S-S deceleration is due to ion caging and plasma electrons effects, and confirms that the underlying mechanism has been correctly identified.

What is insufficient in XMDYN analysis? What does “plasma model” provides?

The hybrid plasma/MD model has a far more favourable scalability in terms of computational time compared to the computationally intensive first-principles approach used in XMDYN, which considers both nuclear and electronic coordinates. The treatment of the electron dynamics using a plasma model allows nuclear dynamics simulations to be performed on complete unit cells and to allow the consideration of the motions of many more nuclei within the unit cells. The use of both approaches and the consistency of their predictions is a significant step towards developing physically realistic and computationally tractable treatments of ionization dynamics within biomolecules and crystalline solids exposed to intense XFEL pulses.

4. page 7. “SFX measurements (which often use 30-50 fs long pulses).” Some studies were done using shorter pulses, <10 fs at SACLA.

We are aware of this; this is why we wrote “often”.

It is enticing to speculate that SACLA data, or in general data collected with shorter pulse durations are less affected by radiation damage. However, in the absence of data demonstrating this we would prefer to refrain from such speculations.

5. page 10. “The restraints for S-S bonds, torsion angles etc. were modified to contain two non-bonded S atoms and two alanine residues, described as alternative conformations to exclude van der Waals interactions”. This sentence is not clear. This treatment may have important effects on the model refinement and should be described in detail.

It is a technical description of the refinement protocol and the constraints used. We tried to make it more clear by writing “To avoid any influence of the normally employed refinement restraints on bond distances, angles and torsion angles, the disulfides were not modelled as normal S-S-bonded cysteines, but as two alanines with two nonbonded S atoms, each described as alternative conformations so as to even exclude van der Waals or other nonbonded interaction restraints.”

We used this refinement protocol for all datasets, including the single pulse data. If this protocol resulted in any systematic errors they would be in all structural models. The single pulse data refine to the same structural models when using either conventional geometry restraints or the “looser” ones described above.

Figure 4. According to this comparison of calculations and experimental data, only 13% of Max Fluence best matches the data. Is it expected?

Both sets of simulations indicate that the atomic motion for medium and high fluence was sufficiently large to disrupt the assumption of periodicity of a sample before the arrival of a

probe pulse. This leads to a gating effect in which information about the positions of the sulphur ions and the structures that they disrupt cannot be determined with any precision. Whatever information can be obtained about the motion of the sulphur ions will, therefore, be biased towards data obtained from the low range of fluence values in which there is minimal disruption of the remaining structure. This view is reflected in the Supplementary Discussion "Comparison of experimental and theoretical results - Effect of the unknown effective fluence distribution" (P. 15 in Supplementary Information).

REVIEWERS' COMMENTS:

Reviewer #2 (Remarks to the Author):

The manuscript has been greatly improved. I think that the paper is now ready for publication.

Reviewer #3 (Remarks to the Author):

The authors have satisfactorily answered all of my comments and questions.